# Research on the Migration and Adsorption Mechanism Applied to Microplastics in Porous Media: A Review

**DOI:** 10.3390/nano14121060

**Published:** 2024-06-20

**Authors:** Lin Zeng, Cong Yuan, Taoyu Xiang, Xiangwei Guan, Li Dai, Dingliang Xu, Danhui Yang, Long Li, Chengcheng Tian

**Affiliations:** 1School of Resources and Environment Engineering, East China University of Science and Technology, Shanghai 200237, China; lynn@stu.cqut.edu.cn (L.Z.); y30211089@mail.ecust.edu.cn (C.Y.); cctian@ecust.edu.cn (C.T.); 2College of Chemistry and Chemical Engineering, Chongqing University of Technology, Chongqing 400054, China; leonzack@stu.cqut.edu.cn (D.X.); lilong20000103@stu.cqut.edu.cn (L.L.); 3College of New Students, Tongji University, Shanghai 200092, China; xiangtaoyu@tongji.edu.cn; 4China Kunlun Contracting and Engineering Corporation (CKCEC), Beijing 100044, China; guanxiangwei@cnpc.com.cn

**Keywords:** microplastic particles, migration mechanism, adsorption, porous media

## Abstract

In recent years, microplastics (MPs) have emerged as a significant environmental pollutant, garnering substantial attention for their migration and transformation behaviors in natural environments. MPs frequently infiltrate natural porous media such as soil, sediment, and rock through various pathways, posing potential threats to ecological systems and human health. Consequently, the migration and adsorption mechanisms applied to MPs in porous media have been extensively studied. This paper aims to elucidate the migration mechanisms of MPs in porous media and their influencing factors through a systematic review. The review encompasses the characteristics of MPs, the physical properties of porous media, and hydrodynamic factors. Additionally, the paper further clarifies the adsorption mechanisms of MPs in porous media to provide theoretical support for understanding their environmental behavior and fate. Furthermore, the current mainstream detection techniques for MPs are reviewed, with an analysis of the advantages, disadvantages, and applications of each technique. Finally, the paper identifies the limitations and shortcomings of current research and envisions future research directions.

## 1. Introduction

Microplastics (MPs), first identified as pollutants in the 1970s [1], have extensive and profound impacts on ecosystems and human health. MPs originate from the degradation of plastic products, synthetic textiles, and industrial discharges, and are widely found in water, soil, and air [2], even entering the food chain. The Chorukh River, the fastest-flowing river in Turkey and among the top ten rivers in the world, has a relatively lagging economy in its watershed. A major challenge in this watershed is estimating the accumulation of pollutants in the sediments of such a fast-flowing river. Nevertheless, the presence of fibers and PE MP was frequently detected in the basin [3]. Recent studies had highlighted significant MP pollution, especially from single-use plastics, in Oman’s mangrove habitats, with plastic waste constituting 73–96% [3]. Additionally, research in Ranchi, Jharkhand, India, found high MP concentrations in indoor air during winter and post-monsoon, with fibrous MPs being more dominant than other forms [4]. Due to their light weight and durability, plastics are widely used in various industries such as cosmetics, industrial manufacturing, food packaging, aerospace, etc. [5], and have become an indispensable and important material. Since the onset of COVID-19, the use of single-use plastics has surged globally, exacerbating the issue of microplastic pollution. According to a 2023 United Nations Environment Programme (UNEP) report, annual global plastic production exceeds 430 million tons, with two-thirds of plastic products designed for short-term use [6]. Plastics are widely used in daily, and global plastic production is projected to reach 33 billion tons by 2050 [7]. As one of the world’s largest producers and consumers of plastics [8], China’s plastic production has been quite large. Plastic waste undergoes a series of natural processes in the environment, such as physical abrasion, chemical oxidation, and biodegradation, which gradually break it down into smaller particles of MPs (particle size less than 5 mm [9]), and the ratio of surface area to volume increases. At microscopic scales, MP particles can readily penetrate fine pores but are also more susceptible to entrapment within narrow pore channels, a characteristic that complicates their migration pathways. Additionally, the adsorption of MPs onto porous media surfaces is a key mechanism for their migration into the environment and carrying toxic substances. Due to the strong adsorption capacity of MPs, they become important carriers of organic pollutants and promote the migration and diffusion of contaminants. The migration of microplastics in porous media may lead to their entry into groundwater and soil systems, contaminating the water quality, thus posing a threat to the safety of drinking water and increasing the difficulty and cost of water treatment. Due to the hydrophobicity and high specific surface area of MPs [10], once they enter the water or soil, they can adsorb a variety of organic pollutants, such as polycyclic aromatic hydrocarbons (PAHs), polychlorinated biphenyls (PCBs), pesticides, heavy metals, and other toxic industrial chemicals. The MPs, as well as the toxic pollutants carried by the MPs, are transferred, accumulated, and enriched in organisms with high nutrient levels through the food chain, which then reach the human body [2,9,11]. Ultimately, phenomena such as biotoxicity and the spread of pathogenic substances can be induced, leading to the disruption of food chain balance and the disturbance of ecological systems. MP contamination can significantly impact soil fertility, agricultural production, wildlife survival, and human health, while soil and sediments, as typical natural porous media, are important sites for MP accumulation and transport, and MPs in soil affect the composition, metabolic processes, and overall functioning of the soil microbiota, which may, in turn, have a serious impact on plant growth, posing a potential risk to the stability and sustainability of agroecosystems [2,9,11,12,13,14]. In addition, MPs are highly persistent in the environment, and once in a porous medium, especially soil, their transport and dispersal may last for decades or even longer.

Research on the migration behavior of microplastics (MPs) in porous media has increased, but fewer studies have systematically investigated the migration mechanisms applied to MPs. This study aims to systematically elucidate the migration mechanisms applied to MPs in porous media and their influencing factors, including the characteristics of MPs, the physical properties of porous media, and hydrodynamic features. Additionally, it explores the adsorption mechanisms applied to MPs during the migration process to provide theoretical support for assessing their distribution and migration in the environment. This paper also reviews current mainstream MP detection techniques, analyzes the advantages and disadvantages of each method and its scope of application, and offers an outlook on future research directions. It is hoped that this study will provide an effective scientific basis and practical methods for environmental pollution control and ecological risk assessment.

## 2. Migration Mechanism of MPs in Porous Media

Particle transport can be traced back as far as the 1960s and 1970s for the study of contaminant transport and migration in groundwater [15]. Porous media consist of a solid skeleton and interconnected pores or voids [16], and these pores give the porous media special permeability and adsorption properties [17]. Compared to normal pollutant particles, MPs with smaller sizes have greater mobility [18], making them more likely to migrate into the environment and be captured by porous media. In addition, the migration of MPs in porous media is a complex and dynamic process, which involves several aspects such as fluid dynamics, particle kinetics, diffusion processes, and pore structure of the media, etc. An in-depth study of these theories and factors can help to better understand the migration mechanism applied to MPs in porous media such as soil, sediment, and filled beds.

### 2.1. Analysis of Flow Field in Porous Media

The flow field serves as the main driving force for the migration of MPs in porous media, and its characteristics, including the flow velocity as well as the flow direction, determine the trajectory and migration rate of MPs.

Within the pores of porous media, fluids usually exhibit laminar flow characteristics accompanied by velocity gradients, which in turn form a shear field. In a homogeneous shear field, MPs (spherical) undergo a rotational motion and are accompanied by the generation of a spherical flow field [19]. When the MPs in the fluid are subjected to different shear stresses, additional energy is provided to the MPs, which contributes to fragmenting into a large number of toxic NPs; meanwhile, the surface energy of the particles is increased, and these particles tend to agglomerate or to be adsorbed into the surroundings [20]. The rotation causes the particles to migrate through the shear field in a path similar, but not identical, to the steering of a spinning ball in flight. When the shear field exhibits inhomogeneity, such as in the pores of a porous medium, the migration of the particles will be subject to a similar but non-uniform and unpredictable deflection. If the MPs are non-spherical, they will be subjected to unbalanced forces that cause them to cross the flow line. The pressure difference generated by the fluid at different locations will also cause the particles to follow the flow line [21]. These complex factors work together to make the MPs finally show random migration motion. The possibility of collision with the surface of the porous medium is increased.

The velocity of the fluid determines the migration rate of MPs in porous media. When the flow velocity in the medium increases, it leads to greater pore flow velocity, which in turn enhances the transfer of particles by the pore flow, and the high flow velocity enhances the resistance and minimizes the settling and trapping of the particles [22]. The increase in fluid flow will also make the shear force suffered by MPs in the migration process more and more significant, substantially shortening the deposition time on the medium’s surface [23]. This heightened shear stress, determined by the fluid’s velocity gradient and viscosity, weakens MPs’ adhesion to the medium, thereby influencing their migration speed and direction. This means that before they reach the surface of the medium, they are carried away from their original position by the fluid or taken deeper into the porous medium.

In Computational Fluid Dynamics Discrete Element Method (CFD-DEM) simulation experiments which were simulated using the properties of MPs, the flow characteristics and particle migration behavior of fluids in porous media at different flow rates were investigated. The pressure drop of the fluid through the porous media bed at different flow rates is illustrated in Figure 1a. When the flow rate increases from 0.1 m/s to 0.3 m/s, the pressure drop increases significantly. Figure 1b shows the particle separation efficiency of each layer (from left to right represents the top to the bottom of the bed) after the porous media bed is meticulously partitioned into 10 layers at a constant flow rate of 0.1 m/s, and the separation efficiency of the first three layers is as high as 38.42%. At a flow rate of 0.1 m/s, particles possess initial kinetic energy and inertia due to the action of the fluid. These particles tend to follow their original trajectory until they reach the topmost layer of the bed (i.e., the first layer), where they collide with the media surface. Upon collision, the particles experience a significant loss of their initial kinetic energy, leading to their interception and capture primarily within the first layer. Consequently, the first layer exhibits the highest separation efficiency. As the fluid penetrates deeper into the bed, the particles are progressively captured, resulting in a decrease in particle concentration. Subsequently, the number of particles intercepted also diminishes, causing a downward trend in the separation efficiency of each successive bed layer. The simulation results show that the increase in fluid flow rate leads to a significant increase in the pressure drop and an enhancement in the momentum and shear stress of the fluid, which in turn affects the particle settling and trapping efficiencies in the porous medium. Specifically, particles are more difficult to settle at high flow velocities, and the possibility of migrating to the bottom of the bed is increased, which changes the distribution of separation efficiency at different levels of the bed, further validating the theoretical assumptions and the validity of the simulation experiments.

### 2.2. KTGF and Surface Interaction Theory

The kinetic theory of granular flow (KTGF) is a theory used to describe the behavior and properties of granular materials during flow. The KTGF model is based on the kinetic theory of granular flow and describes the effect of inter-particle interaction forces on the stress tensor of granular flow, which is similar to the description of molecular thermal motion in the kinetic theory of gases and is used to analyze the solid pressure and stresses generated by the random motion of discrete particles; it takes into account the particle phase, solid pressures, and stresses, as well as the inelastic properties of the particle phase [24]. Weber and Hrenya extended the KTGF approach in their study, and the approach presented in their study opens up the possibility of using the coupled KTGF model to account for particle-particle collisions, agglomeration, and aggregation induced by noncontact forces [25].

The DLVO (Derjaguin–Landau–Verwey–Overbeek) theory is a classical theory describing the interaction forces between colloidal particles. In experimental studies, the interaction energy between MPs and porous media materials (e.g., quartz sand) under different factors can be calculated using the DLVO theory, which in turn explains the migration and deposition behavior of MPs in porous media [26], but the DLVO theoretical calculations are limited in their ability to explain the synergistic translocation behavior of multiple substances [27]. During the migration of MPs, adsorption and surface interactions between particles and porous media surfaces occur, causing MPs to attach to porous media particles or pore surfaces, and the associated forces involved can be categorized into two groups. One group is remote forces, including van der Waals forces and bilayer repulsive forces [28]. Van der Waals forces depend on the chemical composition and structure of the particles and the surface of the medium and can attract or repel particles from the surface of the medium. The double-electric-layer repulsive force is due to the formation of charges on the surface of the particles and the medium, which is enhanced when the particles are in close proximity to the surface of the medium, inhibiting the adsorption of the particles. At the same time, they are the two main interactions considered by the DLVO theory and the main forces for particle adsorption and desorption. The van der Waals attraction (E_vdw_) and double-electric-layer repulsion (E_edl_) can be calculated from Equations (1) and (2), respectively [29,30]:(1)$$E_{\mathrm{vdw}}=-\frac{\mathrm{Ar}}{6h(1\;+\;14h/\lambda )}$$
(2)$$E_{\mathrm{edl}}=\pi \varepsilon_{0}\varepsilon_{r}\;r\{2\xi_{1}\xi_{2}\;\ln\;\frac{1+\exp(-\psi h)}{1-\exp(-\psi h)}+({\xi_{1}}^{2}+{\xi_{2}}^{2})\ln[1-\exp\;(-2\psi h)]\}$$
where A is the Hamaker constant of the MPs, r is the radius of the MPs, *h* is the separation distance (surface to surface) between the MPs and the porous medium, *λ* is the characteristic wavelength of the interaction, often assumed to be 100 nm, *ε*_0_ is the vacuum permittivity, *ε_r_* is the relative permittivity of water, *ξ*_1_ is the surface potential of the MPs, *ξ*_2_ is the surface potential of the porous medium, and *ψ* is the Debye reciprocal length.

According to the DLVO model, the attraction between two microscale MP particles of similar size, caused by van der Waals forces, can be expressed as [31]:(3)$$U_{A}=\frac{\mathrm{aA}}{12\;(R-a)}$$
where U_A_ is the attractive potential energy; a is the particle radius; R is the separation length between the centers of the two particles, and R is significantly smaller than a; and A is Hamaker’s constant.

Another group of forces, called short-range forces, consists mainly of Born repulsive forces and structural or hydration forces, which dominate in the 5 nm range.

The occurrence of adsorption depends on the nature of surface particle interactions. In most of the studies, the combined action of long-range forces has been considered to evaluate the adsorption mechanism. For example, Zhang Y et al. [32] applied the extended DLVO theory (XDLVO) to calculate the interfacial free energy and quantified the adsorption mechanism of neutral organic pollutants to MPs, but other physical adsorption forces such as pore filling, hydrogen bonding, and π–π interactions were not included. Guo Q et al. [33] used the classical DLVO theory based on a hypothetical model to predict the adsorption mechanism of antibiotic resistance plasmids (ARPs) with different kinds of MPs (PS, PE and PVC with a size of 2 μm), where the total interaction energy was calculated as the sum of the bilayer repulsive and van der Waals attractive forces, which was combined with density flooding theory (DFT) to analyze the changes in adsorption capacity. Ibrahim et al. [34] used the classical DLVO theory to calculate the total interaction energy between CMC-nZVI nanoparticles and showed that it can be used to assess the stability of the particles based on the existing solution chemistry conditions.

### 2.3. Modeling of Porous Media with MP Migration

To obtain a deeper understanding of the migration mechanism of particles in porous media, researchers have proposed a variety of porous media models, including the spherical model, the capillaric model, and the constricted tube model. These models are capable of describing the physical structure of the pore distribution of porous media and the interaction between particles and pores.

#### 2.3.1. Spherical Model

Spherical modeling refers to the simplification of media particles with complex geometrical characteristics in porous media into regular equivalent spheres with the same volume or surface area, which are regarded as a collection of spherical collectors, with each unit bed (UBE) containing multiple collectors that do not affect each other in the adherent capture of particles flowing through it. Spherical models include the Happel model, the Kuwabara model, and the Brinkman model, which have been used to study deep filtration in the past [35]. However, the three models are different from each other. Both the Happel and Kuwabara models are based on the Stokes equations and are suitable for describing viscous fluid flow at low Reynolds numbers. The Happel model focuses on uniformly distributed arrays of cylinders [36], while the Kuwabara model addresses randomly distributed parallel cylinders or spheres, introducing a boundary condition where the flow remains unperturbed away from the cylinder or sphere [37]. The Brinkman model extends Darcy’s law by adding a viscous shear term, making it particularly suitable for describing fluid flow at high particle concentrations [38].

#### 2.3.2. Capillaric Model

In the capillary model, the pore structure of a porous medium is simplified to a series of capillary networks because the pore structure of the porous medium is comparable to that of a capillary tube. Conventional models usually assume that these capillaries have a uniform diameter and are parallel to each other, thus simplifying the particle migration process. However, due to the inherent convergent-divergent flow characteristics in porous media, the linear flow behavior assumed by the capillary model is not consistent with the actual situation. Liu, X. et al. [39] innovatively proposed a statistical capillaric model that compares porous structures to curved capillary systems and predicts their performance from core parameters such as porosity, pore size distribution, and curvature. It predicts the permeability and filtration of various porosities more accurately than the conventional spherical model and relates the filtration path to performance. Similarly, H. Cheng et al. [40] developed a generalized theoretical model to calculate relative permeability using fluid distribution properties and a fractal capillary bundle model. Sensitivity analyses of this model showed that tortuosity and pore size distribution have a significant impact on the prediction of relative permeability from capillary pressure profiles. These models enhance the understanding and prediction of MP migration in porous media.

#### 2.3.3. Constricted Tube Model

Compared to the capillary model, the shrinking tube model provides a more realistic perspective for understanding fluid flow and particle migration in porous media. In real porous media, the channel diameter varies with the spatial location, especially in the inter-particle voids, while the channels are narrower in the particle contact points or tightly packed regions. The shrinking tube model treats the pores as dynamically shrinking pipes, where the fluid undergoes continuous contraction and expansion during the flow process, and more accurately describes the effect of the pore structure on the migration of particles. In addition, the shrinking tube model shows higher accuracy than the capillary model in describing the change in channel diameter. MPs generally move through microchannels in sediments, filters, and soils. The constricted tube model simplifies the simulation of MP migration through these narrow channels and aids in predicting their behavior in porous environments by considering factors such as pore size and connectivity. As early as the 1980s, H Pendse et al. [35] used the constricted tube model to study transport processes in particulate media, including momentum transfer, convective mass transfer, axial dispersion, and particle collection and focused on the effect of the wall geometry of the shrinking tube on the model predictions.

### 2.4. Darcy’s Law and the Transport Effect of Brownian Diffusion

Darcy’s law describes the flow law of a fluid in a porous medium, which is a key factor for the migration and distribution of MPs in natural porous media. It correlates the dynamic relationship between flow velocity and media permeability and pressure gradient, thus quantifying the migration process of MPs in porous media and providing a fundamental theoretical framework for modeling the mechanism of fluid (including suspended MPs) movement through such media. Additionally, it is a hallmark for modeling momentum transport through porous media [17]. Historically, Darcy’s law has dramatically changed the analysis of applications such as groundwater flow and has played an important role in the development of the discipline of transport through porous media [41]. Darcy’s law states that the seepage volume *Q* is directly proportional to the difference between the upstream and downstream head (*h*_2_ − *h*_1_) and the cross-sectional area A perpendicular to the direction of the flow, while it is inversely proportional to the seepage length L, namely:(4)$$Q=\frac{KA(h_{2}-h_{1})}{L}$$
where *Q* is the seepage volume per unit time, F is the cross-section, *h* is the total head loss, L is the seepage path length, I = *h*/L is the hydraulic gradient, and K is the infiltration coefficient.

It has been shown that Darcy’s law can be used to analyze the migratory movement of MPs in saturated porous media under specific conditions. Wang et al. [42] used Darcy’s law to calculate the variation of hydraulic conductivity and then analyzed the permeability of MPs with different particle sizes and the clogging mechanism, which proved its applicability in predicting the migration and distribution of MPs in similar environments.

In the micrometer and even nanometer scale range, MP particles are susceptible to significant thermal fluctuations and Brownian motion due to their tiny size [43]. Brownian diffusive motion in MPs, which takes into account the influence of molecular thermal motion of the particles at the microscopic scale, can be interpreted as a combination of Brownian motion and diffusion. In this context, Brownian motion refers to the random motion of MPs in a fluid due to irregular collisions by fluid molecules, while diffusion refers to the process by which particles spontaneously propagate towards a low-concentration region under a concentration gradient due to this motion. This theoretical framework is widely used to describe the motion and diffusion behavior of tiny particles in fluids. It is worth noting that Brownian diffusive motion does not exist in isolation in porous media, and it often works in conjunction with other transport mechanisms (e.g., convection, seepage, etc.), which together determine the migratory behavior of particles in porous media. In the case of one-dimensional steady flow, macroscopic reflection of the migration of suspended particles in saturated, homogeneous porous media is often described by convection–diffusion equations [44,45], namely:(5)$$\frac{\partial \theta C}{\partial t}+\rho \frac{\partial \left(S\right)}{\partial t}=\frac{\partial }{\partial x}\left(\theta D\frac{\partial C}{\partial x}\right)-\frac{\partial qC}{\partial x}$$
where *C* is the colloid concentration in the fluid (mg/L), t is the migration time (min), *ρ* is the density of the porous medium (g/cm^3^), S is the concentration of deposited colloid in the porous medium (mg/g), *θ* is the porosity (-), q is the Darcy flow rate (cm/min), D is the dispersion coefficient (cm^2^/min), and *x* is the distance from the inlet side (cm) [46].

The randomness of Brownian motion leads to the disorder of the movement path of MPs, which enhances the mixing of particles with other components of the medium. Brownian diffusion leads to the uniform distribution of MPs in all directions, which affects the spatial distribution uniformity of particles in porous media. During the migration of MPs, these two mechanisms work together to affect the migration rate and path of the particles. In short, Darcy’s law is used to analyze the flow characteristics of fluids in porous media and predict the migration rate and direction of MPs, while Brownian diffusion increases the uncertainty of particle distribution.

### 2.5. Migration Process of MPs in Porous Media

The migration process of MPs in porous media is a complex and multifactorial dynamic process. This migration process is influenced by a combination of physical, chemical, and biological factors. The physical properties of porous media, such as the size, distribution, and connectivity of pores and pore throats, influence the migration paths and deposition locations of MPs. In nature, porous media have strong non-homogeneity, with abundant interfaces between different components, leading to complex transport processes [47]. On the one hand, large pores and preferential flow may be generated due to permeability differences which are reflected in its pore structure, porosity, pore connectivity, etc., ultimately promote the migration of MPs; on the other hand, the migration of MPs may be impeded by the formation of electrostatic forces between positively charged metal elements on the surface of porous media, such as iron, aluminum, etc., and negatively charged MPs [48], especially because a large amount of metallic elements are often accumulated in natural porous media, such as soil [34]. MPs come in a wide range of sizes, from a few micrometers to several millimeters, and when the particles are only a few micrometers in size, they usually carry a large amount of surface charge, and MPs modified with different surface functional groups will be electrostatically charged due to the difference in surface charge, which can promote or inhibit their transport in porous media [49,50]. W Zhao et al. [27] selected unfunctionalized (MS), microspheres with carboxyl groups (CMS), and sulfonic acid group (SMS) microspheres as model colloidal microplastics to study the effect of surface functional groups on the transport behavior of microplastics. Moreover, they found that the oxygen-containing functional groups such as carboxyl and sulfonic acid groups on the surface of MPs significantly enhanced their electronegativity and hydrophilicity, and these chemical properties effectively weakened the interactions between the microplastic colloids and the medium, and reduced their deposition tendency in the medium, which led to a significant enhancement of the migration ability of the MPs in the porous medium of glass beads. In addition to the functional groups, the presence of DOM can also change the hydrophobicity of MPs [18].

MPs in natural environments are always irregularly shaped, including spherical, fragmented, fibrous, thin films, or foamy, and have strong stability as well as hydrophobicity [10,31,51]. It is well known that some physicochemical properties of MPs will be determined by their compositions (e.g., density, surface hydrophobicity), and these properties have been shown to significantly affect particle migration behavior [10]. It has been shown that the surface hydrophobicity of MPs has a greater effect on the permeation or vertical transport than the density, and the higher the hydrophobicity, the less MPs permeate [18]. Yang X et al. [52] found that MPs with densities close to that of an aqueous medium and those with smaller particle sizes maintain higher dispersibility and lower mobility. Gao J et al. [18] found that MPs with lower density have a stronger buoyancy effect, which hinders their penetration in porous media. When comparing different shapes of MPs, spherical MPs tend to penetrate deeper into porous materials than fragmented and fibrous MPs [10]. In addition, spherical MPs are well dispersed in solution [53]. For fibrous MPs, the diameter of the fibers has a greater effect on the depth of penetration than the length or density of the fibers. Specifically, fibers with finer diameters tend to migrate deeper into porous medium. On the other hand, fragmented MPs (e.g., tire abrasives) do not penetrate as deeply as spherical MPs. It may be due to the entanglement of angular particles within the pore space, which hinders their ability to migrate further [54]. Most of the studied objects have been limited to regular spherical MPs with smooth surfaces, and there are few reports on the migration behavior of fibrous and fragmented MPs.

The rich types and complex compositions of porous media in nature, such as the biological activities of microorganisms in soil, may also affect the migration process of MPs in porous media. For example, the size and surface properties (e.g., zeta potential, surface roughness, hydrophobicity, and surface chemistry) of MPs change as a result of bacterial attachment to MPs [55,56]. He L et al. [57] found that bacteria adsorbed on quartz sand formed larger-sized CMP (negative carboxylate-modified MP)–bacterial clusters and additional deposition sites, resulting in a reduced transport of CMPs and enhanced their deposition in the sand column. In addition, soils are usually filled with large amounts of wastewater containing surfactants. The study by Jiang et al. [58] clarified the relationship between the area occupied by surfactants on the surface of MPs and the migration of MPs and demonstrated that the presence of surfactants could effectively increase the migration of microplastics.

### 2.6. Filter Separation Material of Porous Media

The selection of filter media will directly affect the accuracy and reliability of the results of MP migration experiments. Filter media can be divided into natural mineral filter media, metal filter media, ceramic filter media, fiber filter media, and nanomaterial filter media. The column test is one of the common methods used by researchers to simulate the migration behavior of MPs in porous media, and glass beads, quartz sand, or captured soil are often used as filler materials [10]. By choosing appropriate filter media, the migration mechanism of MPs under different environmental conditions can be simulated and investigated, providing important experimental data for MP pollution control. In addition, the combination of column experiments, numerical simulations, and microscopic imaging techniques is of profound significance and necessity for the study of the migration mechanism of MPs in porous media [59].

#### 2.6.1. Hydrophilicity and Hydrophobicity

Hydrophilic media surfaces are more adsorptive for water, which may reduce the adsorption of MPs on the media. In particular, those MPs with hydrophilic functional groups, such as CMS and SMS, increase the hydrophilicity of colloidal microplastics and reduce their adhesion to the surface of hydrophilic media, which facilitates the transport of colloidal microplastics in the media [27]. In contrast, hydrophobic media are more likely to adsorb hydrophobic MPs, which leads to the retention of MPs in the media. In addition, hydrophilic media (e.g., quartz sand) are more likely to adsorb particles of polarized or charged MPs via electrostatic adsorption and hydrogen bonding, whereas hydrophobic media (e.g., hydrophobically treated glass beads) may repel these particles.

#### 2.6.2. Absorption Capacity

Adsorption capacity is a key factor in the role of porous media materials in filtration and separation processes. Materials with high adsorption capacity (e.g., activated carbon, zeolite) are more effective in capturing and removing small particles. Activated carbon is ideal for simulating particle migration in natural soil environments due to its extremely high specific surface area and porous structure. Wood, as a natural biomass, exhibits a multi-scale porous structure ranging from macro to nano levels [60], and the adsorption capacity of wood can be significantly improved by making activated carbon from wood through high-temperature carbonization treatment for use in column experiments. This is due to the fact that its porous structure and surface functional groups contribute to the adsorption efficiency [61]. Zeolites have a high specific surface area and abundant pore structure [62], enabling them to adsorb a large number of molecules and improve separation efficiency. Therefore, it is also able to play an important role in a variety of column tests, especially in adsorption and separation applications.

#### 2.6.3. Environmental Friendliness

Environmental friendliness is one of the important considerations when selecting filter media in the experiment. Biodegradability, regeneration ability, and potential environmental impact of a material are all key indicators in assessing its environmental friendliness. Natural quartz sand and modified soil materials are widely used because of their low cost, low environmental impact, and good regenerative properties. In addition, glass beads are often used as a simulation material for natural sediments and are often used in studies to assess the migration behavior of MPs in sediments due to their similar physical and chemical properties [54]. These materials can be naturally degraded or recycled after use and are relatively harmless to the natural environment, thus reducing the environmental burden.

## 3. Detection Techniques for MPs in Porous Media

Microplastics show great diversity in size, morphology, chemical composition, and structural features, which poses a great challenge for their detection. In recent years, MP detection techniques have been significantly developed, as shown in Figure 2. For the detection of MPs in porous media, the commonly used methods include visualization, FTIR, RS, SET, TEM, and CT scanning techniques. These techniques are not only able to effectively detect and analyze the presence and migration of MPs in porous media but also significantly improve the accuracy of the analysis, providing important data support for MP contamination studies. However, there is still no universal, efficient, fast, and low-cost detection and analysis method [59], so to more comprehensively understand the migration behavior and influence mechanism of MPs in porous media, it is still necessary to continuously improve and refine the detection techniques in the future.

### 3.1. Visual Method

The most widely used method is visualization, which is a simple and easy identification technique [24,63]. MPs can be readily identified either visually by the naked eye or through the use of a microscope, which enables the precise determination of their shape, size, color, as well as other critical physical attributes, including transparency, surface characteristics, and aggregation status. Since the particle size of MPs that can be observed by the naked eye is generally 1–5 mm, and the detection of MPs using a microscope has limitations, the visual inspection method is not highly reliable. Their errors tend to increase with decreasing microplastic particle size [25], with error rates ranging from 20% [64] to 70% [65]. Although visual inspection is generally not used as a stand-alone identification method, its low cost and ease of use, in combination with other more accurate analytical methods such as Raman spectroscopy, Fourier transform infrared spectroscopy (FTIR), and scanning electron microscopy (SEM), can provide a more comprehensive analysis of the type, content, and morphological characteristics of MPs in samples, and thus a more accurate assessment of the contamination of MPs in the environment. Therefore, it remains a common method for analyzing and identifying MPs.

### 3.2. Scanning Electron Microscopy (SEM) and Transmission Electron Microscopy (TEM)

Compared to optical microscopy, electron microscopy offers significantly higher resolution and magnification. SEM generates high-resolution images by scanning the electron beam on the sample surface and utilizing signals interacting with the sample surface (e.g., secondary electrons, back-scattered electrons, Roche electrons, etc.), which enables detailed observation of microscopic details such as the morphology, surface features, and dimensions of MPs. In addition, it is possible to image the surface of porous media at the microscopic scale. SEM is often used in conjunction with an energy spectrometer (e.g., EDS) to generate images of the surface morphology and elemental composition of MPs, which can be used to further characterize the chemical composition of MPs [66,67], thus providing more comprehensive information. The combination of SEM-EDS and optical microscopy enables rapid and efficient screening of large quantities of microplastic particles, thereby reducing the likelihood of misidentification. SEM-EDS has been demonstrated to enhance the accuracy of test results through chemical analysis compared to the use of SEM alone. For example, Jiang H et al. [68] used SEM-EDS analysis to assess the morphology and elemental composition of MPs before and after incubation with sediments when investigating the hydrophilicity mechanism of MPs. However, the technique necessitates complex sample preparation procedures, which not only increase the assay time but also require the use of expensive instruments [63].

TEM is imaging by transmitting a beam of electrons through the interior of a sample. The electron beam passes through the sample, where some of the electrons are scattered or absorbed, while others are transmitted to a detector to produce a two-dimensional image of the internal structure of the sample. It combines inverse spatial diffraction, real-space imaging, and spectroscopic techniques for comprehensive characterization in the domains of time, space, momentum, and increasingly, energy, with excellent resolution [69]. SEM typically has a low resolution, generally in the nanometer to sub-nanometer range, while TEM, on the other hand, provides high-resolution images [47] that can reach sub-nanometer to atomic levels. The current literature reports that TEM can resolve atomic spacings down to 0.039 nm [70]. SEM is able to look at the overall morphology of the sample and the texture of the surface, while TEM is able to provide detailed structural information about the interior of the sample.

### 3.3. Raman Spectroscopy (RS) and Fourier Infrared Spectroscopy (FTIR)

In vibrational spectroscopy (VS), RS and FTIR are the most commonly used analytical method techniques in the study of MPs [71], and both techniques offer multiple advantages, such as non-destructiveness, low sample requirements, potential for efficient screening, and environmental friendliness [72]. However, traditional FTIR and Raman analyses are limited by light diffraction, and it is difficult for the spatial resolution of MP detection to reach the nanometer scale [73].

Raman spectroscopy is the method of choice for characterizing small microplastics (<20 μm) and is a vibrational spectroscopy technique based on inelastic scattering of light [72]. It is based on the analysis of polymer types by backscattered light at different frequencies from laser beams with different molecular and atomic structural features, so that the characteristic Raman spectra in each polymer can be obtained to achieve high-throughput detection of MPs with a resolution higher than that of FTIR [74], but it also results in a significantly higher imaging runtime than that of FTIR [75]. RS functions similarly to a chemical fingerprint, allowing for the identification of components within a sample. Additionally, it facilitates the quantitative analysis of microplastics using an RS microscope. A significant advantage of RS is its excellent spatial resolution.

Theoretically, Raman spectroscopy has a spatial resolution of 1 μm and can detect MPs as small as 1 μm in size [76]. However, Raman spectroscopy cannot readily identify pigmented fibers or particles and is sensitive to additives and pigments, thus affecting the determination of plastic-type [77,78]. Raman spectroscopy has an inherently low signal-to-noise ratio, and the use of a laser can cause sample heating, leading to background emission and potential polymer degradation [72]. Although Raman microspectroscopy is a reliable method for identifying microplastic properties, it is costly and ineffective for quantifying the mass content of microplastics in soil. This limitation is primarily due to the autofluorescence of soil organic matter (SOM), which significantly hampers the applicability of RS in soil analysis [78,79,80]. Recent studies have shown that combining Raman spectroscopy with machine learning (ML) is an effective method for recognizing and classifying MPs, especially when they are found in environmental media or mixed with various types [81].

The FTIR technique is based on the measurement of molecular absorption eigenfrequencies to determine the type of MPs in a sample by comparing the sample spectra with standard spectra; it has a greater ability to identify polar groups [74] and can usually detect MPs with particle sizes larger than 20 μm [76], while it is not suitable for analyzing NPs [82]. Different microplastics exhibit distinct infrared spectra. When scanned with infrared light, their unique fingerprints and positional information are recorded, allowing the distribution of microplastics to be mapped and visualized based on infrared intensity. Detecting and analyzing individual microplastics under a probe is time-consuming and laborious. However, FTIR analysis can quickly and accurately identify microplastic particles without interference from fluorescence [83]. Conventional FTIR techniques are used to detect plastic fragments or visible MPs, whereas combining FTIR with microscopy (μFTIR) allows the detection of MPs up to 10 μm in size [78]. FPAs (focal plane arrays) are detectors used for optical imaging that read the optical signals of multiple pixels simultaneously without the need for mechanical scanning. When FPAs are combined with micro-Fourier transform infrared spectroscopy (micro-FTIR), multiple spectra of MPs on a surface can be collected simultaneously, which in turn allows for fast and accurate identification of MPs in environmental samples [78,84]. Zhang Z et al. [85] developed an innovative multispectral coupling method combining μ-FTIR and u-Raman techniques, enabling efficient one-stop detection of MPs. This method overcomes traditional single-spectrum method limitations, such as particle size constraints and fluorescence interference, while significantly reducing sample pretreatment costs and promoting environmental sustainability. Additionally, it demonstrates excellent applicability to both environmental and biological samples.

### 3.4. Magnetic Resonance Imaging (MRI)

Magnetic resonance imaging (MRI) is a non-destructive, non-invasive imaging technique developed based on the principle of NMR, widely used for non-invasive imaging of fluid flow in various porous media and measuring certain hydrocarbons [86,87]. MRI can also provide local average porosity and pore size spatial distribution information [88]. It utilizes atomic nuclei to generate resonant signals under the action of static magnetic and radio frequency fields and generates three-dimensional structural images through demodulation, spatial coding, and image reconstruction. Typically, MRI spatial resolution ranges from 0.5 to 1 mm [89], and its high resolution can reach the sub-millimeter level [90]. High-field MRI studies achieve a level of resolution comparable to that of CT (in-plane accuracy of approximately 0.4 mm), while eliminating the need to rely on contrast agents or exposure to ionizing radiation [89]. However, higher resolution often decreases the signal-to-noise ratio (SNR) and contrast-to-noise ratio (CNR), compromising image readability and accuracy when noise levels become significant [89]. Image quality is largely dependent on SNR and material contrast differences within the imaging object [91]. Since the early 21st century, MRI has been used alongside velocimetry to observe fine-particle deposition within porous media, exploring hydrodynamic properties and fluid transport [92]. For example, Paulsen J L et al. [93] used MRI to probe the structure and flow velocity within the inter-particle space of an agar-bead-stacked bed under water-saturated conditions and utilized image processing techniques to obtain the internal structure of the bed and determine the pore parameters of the porous medium. Paramagnetic tracers have also been utilized in MRI to track heavy metal pollutants [47], indicating the potential for combining tracers and MRI to study MP migration in porous media. Furthermore, combining MRI with techniques like micro-CT (μCT) offers comprehensive multimodal detection. Węglarz et al. [90] successfully mapped the 3D distribution of localized water in non-uniform rocks, integrating MRI data with high-resolution μCT structural images.

### 3.5. CT Scanning Technology

CT scanning technology is a non-destructive and very effective imaging technique for visualizing the internal characteristics of solid objects by acquiring digital images of the internal geometry and mechanical properties of the object, providing hundreds of scanned images through the thickness of the specimen [94]. The principle is to use X-rays passing through the object and recorded by a detector; the X-ray source and detector are rotated 360° around the object to be tested, a large amount of X-ray projection data is collected through different angles, the detector records the amount of X-rays absorbed through the object, and information about the different areas is obtained based on the differences in absorption. The collected X-ray projection data are reconstructed into a three-dimensional grayscale image by a computer. Three-dimensional grayscale images are produced by different linear absorption coefficients of colloidal materials, porous media, and carrier fluids [95]. CT scanning techniques are usually performed in seconds to minutes, which is faster than other imaging techniques such as magnetic resonance imaging (MRI). However, they are expensive and have limited availability [94] and require additional spatial measurement conditions [96] to ensure the safe operation of the equipment and accuracy of the imaging quality. From the CT images, the features of porous media can be extracted in detail, including the morphological dimensions of pores, particles, and cracks and their spatial distribution, internal connectivity, and the degree of zigzagging of the paths, which is conducive to the evaluation of the physical properties of the porous media like diffusion performance and permeability.

In the field of CT scanning technology, micro-CT (μCT) has emerged in the field of imaging and analyzing samples at the microscopic scale due to its excellent spatial resolution. μCT, with resolutions up to the micron or even submicron level, is capable of accurately capturing fine details inside tiny samples or complex structures. Since around the middle of the first decade of the 2000s, μCT has become a major imaging tool in experimental studies of DBF processes [97,98,99]. Recent studies have shown that an innovative combination of μCT and magnetic resonance velocimetry (MRV) can comprehensively analyze the complex migration processes of particles along porous media, imaging porous media and particle deposits using high-resolution μCT that captures particle deposition sites at discrete time steps and quantitatively measuring pore fluid flow using MRV [95,100]. Blurred, low-resolution CT images of porous media can obscure structural details and hinder the analysis of transmission processes. To address this, Cai Z et al. [101] developed a novel super-resolution generative adversarial network (HRGAN) using high-resolution representation learning. Their study demonstrated that HRGAN effectively enhances resolution and reduces blurring in μ-CT images. Compared to μCT, nano-CT offers higher resolution (10–20 nm) [47] and better sensitivity, though it has a slightly lower imaging speed. Therefore, the choice between CT scanning technologies should be based on the specific resolution and application requirements.

Considering the advantages and disadvantages of each technique, as shown in Table 1, FTIR and RS remain effective for detecting microplastics in porous media due to their ability to provide chemical composition information, aiding in the differentiation of plastic types. FTIR, in particular, is suitable due to its high sensitivity and low interference. SEM combined with EDS offers valuable morphological and elemental composition information. MRI and μCT are useful for observing the 3D structure of porous media and identifying microplastic locations and distributions. For studying MP migration paths, CT scanning is ideal as it provides 3D images for analysis, while techniques like SEM and FTIR can complement this by providing chemical information.

## 4. Conclusions

This paper reviews the migration mechanism of MPs in porous media, analyzes the influencing factors, and discusses the adsorption mechanism during the migration process. Meanwhile, the current mainstream MP detection techniques, such as RS, FTIR, SEM, and CT scanning techniques, are outlined. However, each technique has its limitations. Existing studies are still insufficiently in-depth on the migration mechanism of MPs; in particular, the migration behaviors of MPs in fibrous and fragmented forms are less reported. Comprehensive and detailed studies are urgently needed. Additionally, there is a lack of systematic analysis on the effect of coexisting substances on MPs’ mobility in porous media. The complexity of environmental porous media, such as metallic minerals on soil surfaces, further complicates research difficulty. To improve the accuracy and reliability of research data, experiments are encouraged to utilize realistic porous media, such as natural sediments, as filling materials. Future research should develop advanced mathematical models to enhance study accuracy and construct an integrated research system including column experiments, numerical simulations, and imaging techniques to more accurately predict the migration behavior of MPs under varying factors. Additionally, it is crucial to develop efficient, fast, and cost-effective MP detection techniques. Future research is expected to accurately reveal the migration patterns of MPs, providing a scientific basis and technical support for preventing and controlling MP pollution.

## Figures and Tables

**Figure 1 nanomaterials-14-01060-f001:**
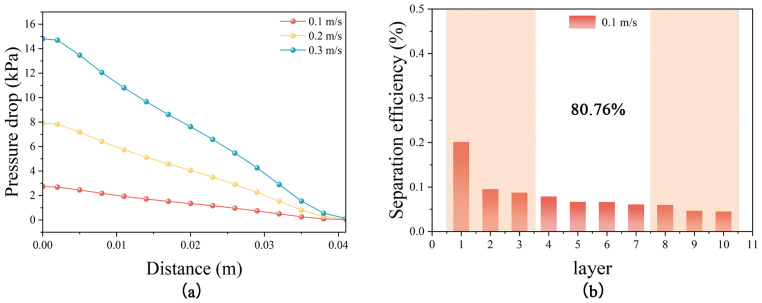
(**a**) Pressure drop corresponding to different flow rates in porous medium; (**b**) separation efficiency of each unit bed. (The size of MPs in the simulation is 100 μm).

**Figure 2 nanomaterials-14-01060-f002:**
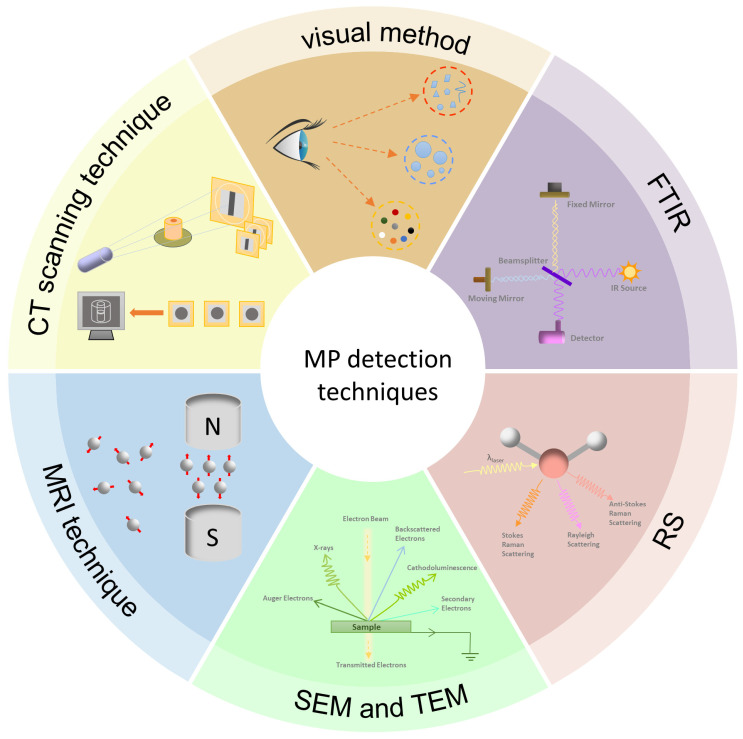
MP detection techniques.

**Table 1 nanomaterials-14-01060-t001:** Comparison of advantages and disadvantages of MP detection techniques.

Title 1	Advantages	Disadvantages	Resolution
Visual Method	Simple to operate, lower cost	Inefficient, time-consuming, highly subjective, limited accuracy	1–5 mm
SEM	High spatial resolution, 3D morphological information, can be combined with EDS analysis	Complex sample handling, high equipment cost	Nanometer to sub-nanometer level
TEM	Extremely high spatial resolution.	Complex sample handling, high equipment cost	Sub-nanometer to atomic level
FTIR	Non-destructive testing, fast, suitable for batch analysis	High sample requirements, time-consuming	20 μm
RS	Non-destructive testing, no pre-treatment, direct analysis, high chemical specificity	Requirements for particle size, fluorescence interference, high cost of equipment	1 μm
MRI	Non-destructive testing, high accuracy	High equipment cost, complicated operation, limited image quality	0.5–1 mm
μCT	Non-destructive inspection, providing high-resolution 3D imaging	High equipment cost, limited resolution, additional spatial inspection conditions	1–5 μm

## Data Availability

Data are contained within the article.

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
