# Peer review of "Research on the Migration and Adsorption Mechanism Applied to Microplastics in Porous Media: A Review"

_nanomaterials, 2024, doi:10.3390/nano14121060_

Round 1

Reviewer 1 Report

Comments and Suggestions for Authors

The manuscript “Study on the Migration and Adsorption Mechanism of MPs in Porous Media: A Review“ by  Lin Zeng et al. deals with the investigation of mechanism of adsorption and migration of microplastics into porous media. Additionally, some MPs detection and identification tehniques are revealed. This topic is of high importance due to the significant environmental and health impacts associated with microplastic contamination. The entire manuscript needs a comprehensive reconsideration of the content, and clarity to ensure it meets the highest standards of academic and scientific rigours.

Some of the findings are shown below.

Abstract: “detection technologies for MPs” is not correctly.

The Introduction section is not well organized. The literature review should be improved. The main objectives of this research and the significance of the study were not highlighted.

The affirmation from Line 35-36 needs a reference.

The data from Line 42-43 must be supported by a reference. How the authors interpret this data with the affirmation from the Line 35-38?

Some phrases are not clear, and hard to understand, for example: “And the behavioral properties at such tiny scales make the dynamic behavior of MPs particles in the environment more complex and difficult to predict.”

“The MPs, as well as the toxic pollutants carried by the MPs, are transferred, accumulated, and enriched in organisms with high nutrient levels through the food chain, which then reach the human body, ultimately leading to biological”.

Line 111 – the abbreviation “CFD-DEM” should be explicit since it was used first time.

Eq (1) and (3): the same constant is noted in two distinct ways: A and AH.

3. Detection Techniques for MPs in Porous Media: FT-IR, RS, SET, TEM, and CT scanning techniques – please detail them.

Figure 2 – “technology” should be replaced with “technique”

Line 380: What kind of “other physical properties” are the authors referring to?

The authors should expand the discussions for both Chapter 2 and 3, about the migration and adsorption mechanisms of MPs as well the detection and characterization of MPs in soil and sediment media.

Reviewer 2 Report

Comments and Suggestions for Authors

The paper shows that migration and adsorption mechanism of MPs in porous media. The mechanisms about migration and adsorption mechanism of MPs in porous media were discussed by several technologies. The studies were quite systematic and the resulted were well organized by the authors. I’d like to recommend the publication of this nanomaterials after revision.

(i) In Figure 1 (b), the best separation efficiency for 1 layer under 0.1 m/s should be explain about its main reason.

(ii) In section 3, Figure order should be checked in the context. The advantage and disadvantage of different technology should be provided and discussed in a table.

(iii) In MPs detection technology, the suitable technology for MP detection should be discussed in the context.

(iv) Among different detection technology, resolution should be discussed in the context.

Round 2

Reviewer 1 Report

Comments and Suggestions for Authors

The paper was improved and I recommend its publishing.